# The Important Role of Protein Kinases in the p53 Sestrin Signaling Pathway

**DOI:** 10.3390/cancers15225390

**Published:** 2023-11-13

**Authors:** Karsten Gülow, Deniz Tümen, Claudia Kunst

**Affiliations:** Department of Internal Medicine I, Gastroenterology, Hepatology, Endocrinology, Rheumatology and Infectious Diseases, University Hospital Regensburg, 93053 Regensburg, Germany; deniz.tuemen@ukr.de (D.T.); claudia.kunst@ukr.de (C.K.)

**Keywords:** p53, Sestrins, stress response, mTORC, nuclear factor erythroid 2-related factor 2 (Nrf2), mitophagy, autophagy

## Abstract

**Simple Summary:**

Cells experience various stress conditions, including replicative stress, oxidative stress, toxin-induced damage, and pathogen exposure. The tumor suppressor p53 initiates intricate cellular responses to mitigate such stress. Sestrins, induced by p53, play a pivotal role in these responses. The Sestrin family has three members, with Sestrin 2 being the most extensively studied. Here, all three Sestrins are considered, with a special focus on Sestrin 2. Sestrins trigger complex signaling cascades involving kinases and kinase complexes such as mTORC, AMP-activated kinase (AMPK), and Unc-51-like protein kinase 1 (ULK1). These kinase-driven responses enable cells to defend against cellular stress, facilitate repairs, and adapt to changing conditions, preventing damage accumulation in macromolecules such as nucleic acids, proteins, and lipids, and thereby averting malignant transformation. Thus, Sestrins essentially contribute to the prevention of cancer development. Because Sestrins also support adaptation to cellular stress and can thereby promote cell survival, these stress responses can also protect malignant tumor cells. Therefore, such anti-stress responses are a double-edged sword—on the one hand they prevent the onset of neoplasia, but on the other hand they can also shield an already established malignancy from the induction of cell death.

**Abstract:**

p53, a crucial tumor suppressor and transcription factor, plays a central role in the maintenance of genomic stability and the orchestration of cellular responses such as apoptosis, cell cycle arrest, and DNA repair in the face of various stresses. Sestrins, a group of evolutionarily conserved proteins, serve as pivotal mediators connecting p53 to kinase-regulated anti-stress responses, with Sestrin 2 being the most extensively studied member of this protein family. These responses involve the downregulation of cell proliferation, adaptation to shifts in nutrient availability, enhancement of antioxidant defenses, promotion of autophagy/mitophagy, and the clearing of misfolded proteins. Inhibition of the mTORC1 complex by Sestrins reduces cellular proliferation, while Sestrin-dependent activation of AMP-activated kinase (AMPK) and mTORC2 supports metabolic adaptation. Furthermore, Sestrin-induced AMPK and Unc-51-like protein kinase 1 (ULK1) activation regulates autophagy/mitophagy, facilitating the removal of damaged organelles. Moreover, AMPK and ULK1 are involved in adaptation to changing metabolic conditions. ULK1 stabilizes nuclear factor erythroid 2-related factor 2 (Nrf2), thereby activating antioxidative defenses. An understanding of the intricate network involving p53, Sestrins, and kinases holds significant potential for targeted therapeutic interventions, particularly in pathologies like cancer, where the regulatory pathways governed by p53 are often disrupted.

## 1. Introduction

Protein kinases serve as pivotal regulators within cellular signaling networks [1]. Through the phosphorylation of target proteins, they exert precise control over their activity, subcellular localization, and function, thereby governing a wide array of cellular processes. Therefore, the catalytic activity of kinases is intricately modulated by various regulatory proteins, including other kinases, giving rise to multilayered protein kinase cascades. Additionally, the activity of kinases can be modulated in response to extracellular cues such as second messengers or interactions with cell surface receptors and ligands. A further layer of regulation involves the dephosphorylation of proteins through the action of protein phosphatases, which functionally counteract kinases.

The perturbation of kinase activity can lead to profound alterations in cellular signaling, often resulting in deregulated cellular behavior. Consequently, it is not surprising that dysregulated kinases frequently exhibit oncogenic properties and play pivotal roles in promoting both the survival and proliferation of cancer cells [2,3,4].

There are two main types of protein kinases: (i) serine/threonine kinases and (ii) tyrosine kinases [1]. 

(i)Serine/threonine kinases are pivotal components of cellular signaling networks. These kinases catalyze the phosphorylation of hydroxyl groups (OH groups) attached to the amino acid residues serine and threonine. Their activity is typically modulated by second messengers (e.g., cyclic adenosine monophosphate (cAMP) or cyclic guanosine monophosphate (cGMP), as well as 1,2-diacylglycerol (DAG), calcium ions (Ca^2+^), calmodulin, phosphatidylinositol-3,4,5-trisphosphate (PIP3), and various derivatives of phospholipids). Serine/threonine kinases are critical components in cellular signaling networks, functioning as molecular switches that transmit extracellular signals and govern intracellular responses. Consequently, they are central to comprehending cell physiology and represent potential therapeutic targets for diseases associated with disrupted signaling pathways [1,5].(ii)Tyrosine kinases can be classified into two groups: membrane-bound tyrosine kinases (e.g., KIT (CD117), epidermal growth factor (EGF) receptor, insulin receptor, human epidermal growth factor receptor 2 (HER2)) and non-membrane-bound tyrosine kinases (e.g., Abelson murine leukemia viral oncogene homolog 1 (ABL1), SRC, Janus kinases). Membrane-bound tyrosine kinases include receptors with intrinsic kinase activity (where the kinase is an integral part of the receptor) and receptors with associated kinase activity (where the kinase binds to the receptor). Upon ligand binding to a membrane-bound or non-membrane-bound tyrosine kinase, conformational changes occur, promoting the formation of homodimers or heterodimers. This process triggers the autophosphorylation of specific tyrosine residues on the kinase, subsequently enabling the recruitment of additional proteins/kinases and initiating intricate signaling cascades [1,2].

Precise control of protein phosphorylation states is essential for modulating protein function, localization, and signaling, all of which are crucial for cellular homeostasis. The phosphorylation status of a protein is controlled by the finely regulated functions of kinases (phosphorylation) and phosphatases (dephosphorylation). Dysregulation of these processes can lead to diseases, also highlighting the pivotal role of phosphatases in cellular homeostasis [6]. Taken together, protein kinases represent essential regulators in intracellular signaling and by phosphorylation of the hydroxyl (OH) groups of specific amino acids, they induce complex signal cascades. However, a deregulated protein phosphorylation can contribute to tissue damage or the development of malignancies.

In this review article, we summarize the role of kinases in p53/Sestrin-regulated stress responses. Tumor suppressor p53 is induced during cellular stress, and among its target genes are Sestrin 1, 2, and 3. These Sestrins can induce various protective responses, often relying on the activation or inhibition of multiple kinases. We explain the role of these kinases and provide an overview of the role of Sestrins in cancer.

### 1.1. p53—Tumor Suppression and Beyond

p53 is a tumor suppressor protein activated by cellular stressors such as ionizing radiation, hypoxia, carcinogens, and oxidative stress. Upon activation, p53 initiates cell-cycle arrest, promotes DNA repair, or induces apoptosis via multiple pathways [7,8]. Besides p53, p63 and p73 complete the family of p53 transcription factors and exhibit similar and distinct functions from p53. The protein structures of the three family members display a high degree of similarity due to their modular organization. The three main domains, namely the transactivation domain (TAD), DNA-binding domain (DBD), and oligomerization domain (OD), are characteristic features typical of transcription factors and are highly conserved [9,10,11,12,13]. The p53 family encompasses over 40 isoforms arising from various promoters and alternative splicing processes [13,14,15].

p53 is encoded by the *TP53* gene located on the short arm of chromosome 17. The gene comprises 11 exons, resulting in a 43.7 kDa protein. *TP53* encompasses three promoters—two upstream of exon 1 for the transcription of full-length p53 (FLp53) and one promoter within the intronic sequence between exon 4 and 5 for the expression of Δ133p53 [16,17]. Alternative splicing of intron 9 generates three additional isoforms of full-length p53α, p53β and p53γ. Both p53β and p53γ lack the oligomerization domain (OD). Recent investigations have elucidated the existence of an alternative 3′ splice site situated within intron 6, which gives rise to the hitherto uncharacterized isoform p53Ψ. At the molecular level, p53Ψ is distinguished by the absence of substantial parts of the DNA-binding domain, the nuclear localization signal, and the tetramerization domain, which represent essential components of full-length p53 [18].

The N-terminal truncated isoforms Δ40p53 result from alternative splicing of exon 2 and initiation of translation at an ATG start codon at position 40. It lacks a portion of the N-terminal transactivation domain (TAD). Δ160p53 isoforms, which lack amino acids 1–159, arise from translation initiation at an ATG start codon at position 160. To date, a comprehensive repertoire of 13 distinct p53 isoforms has been elucidated, encompassing p53, p53β, p53γ, p53Ψ, Δ40p53α, Δ40p53β, Δ40p53γ, Δ133p53α, Δ133p53β, Δ133p53γ, Δ160p53α, Δ160p53β, and Δ160p53γ [19,20,21,22]. For instance, the full-length isoforms are responsible for transcribing downstream genes related to cell cycle arrest, apoptosis, and metabolic regulation, and higher Δ40p53:FLp53 levels are associated with enhanced apoptosis and reduced tumor recurrence rates [23]. In contrast, Δ133p53 and Δ160p53 isoforms are recognized for their pro-survival and anti-apoptotic capabilities [17,24,25,26,27]. These various functions of p53 can only be explained by this multitude of isoforms. p53 is a cellular stress protection factor. By the induction of cell cycle arrest, generally initiated by FLp53, cells gain time to repair damages (e.g., DNA damage) and adapt to specific cellular stress conditions. If the cell fails in damage repair or stress adaptation, cell death is induced through the Δ40p53 isoforms. Thus, p53 protects against the accumulation of cell damage and, consequently, the development of oncogenic changes. Therefore, p53 is one of the most important regulators of stress responses. Cellular stress arises from various factors, such as UV radiation, ionizing radiation, hypoxia, toxins, oxidative stress, replicative stress, and acquisition of mutations [14,28,29,30]. In addition to its role as a regulator of stress responses, p53 also monitors genomic stability by inducing a DNA repair response. p53 also controls the cell cycle and metabolism. It is one of the most well-known regulators of intrinsic apoptosis. Therefore, p53 serves as a protective shield, preventing mutations and eliminating aberrant cells [7,13,31,32,33,34].

Since cellular stress can arise from different sources, there is also a range of stress response proteins that, alone or in interaction with others, restore cellular homeostasis. Sestrins are one group of such stress response proteins, and their regulation is closely linked to p53.

### 1.2. Sestrins—The Universal Shield against Cellular Stress

The name “Sestrin” is derived from the term “SEnsor of STRess signals” combined with the suffix “-in”, which is often used in biology to denote a protein. The name reflects the primary function of Sestrins as sensors within cells that are used to detect and respond to various forms of stress. Sestrins are a family of highly conserved proteins initially discovered in the early 2000s and play a crucial role in cellular stress responses and regulation of the metabolism [35,36,37,38]. Sestrins are evolutionarily conserved proteins found in a wide range of organisms, from yeast to humans, indicating their importance. The Sestrin family consists of three members: Sestrin 1, Sestrin 2, and Sestrin 3, with Sestrin 2 being the most extensively studied [39]. Sestrin 1 exhibits widespread tissue expression, including the pancreas, kidney, skeletal muscle, lung, and brain, and becomes activated in response to oxidative and irradiation stresses in a p53-dependent manner [40,41]. The induction of Sestrin 1 is directly associated with the regulation of cell proliferation [37]. Sestrin 2, discovered via microarray analysis in glioblastoma A172 cells under hypoxia-inducible factor 1 (HIF1)-independent hypoxic conditions, is situated on chromosome 1p35.3. It is activated by oxidative stress-induced DNA damage and over-nutrition stress in the lung, liver, adipose tissue, kidney, and pancreas, following a p53-mediated pathway [36,38,41,42]. Sestrin 3, initially identified by in silico database analysis, is located on chromosome 11q21. It is mainly activated by FoxO1 and FoxO3. However, there is evidence that Sestrin 3 can also be regulated by p53. It is expressed in skeletal muscle, intestine, liver, adipose tissue, kidney, colon, and the brain [35,41,43]. Sestrin 3 exhibits some functional redundancy with Sestrin 2 in promoting metabolic processes and in autophagy induction in liver and colon tissues [41].

Taken together, Sestrins are known for regulating various cellular processes, including antioxidant defense, DNA repair, autophagy/mitophagy (a process that removes damaged cellular components), and metabolism. They act as p53-controlled regulators that help cells adapt to different cellular stressors (Figure 1).

These anti-stress responses involve two cellular complexes, mTORC1 and mTORC2. In both complexes, the catalytic subunit and central serine/threonine kinase is the target of rapamycin (mTOR). In the mTORC1 complex, the regulatory-associated protein of mTOR (Raptor) serves as a scaffold protein that helps bind substrates and regulatory proteins to mTORC1. The mammalian lethal with SEC13 protein 8 (mLST8) stabilizes the complex, while the proline-rich Akt substrate 40 kDa (PRAS40) and DEP domain-containing mTOR-interacting protein (DEPTOR) regulate the kinase activity of mTOR. In the mTORC2 complex, the rapamycin-insensitive companion of mTOR (Rictor) functions as a scaffold protein, and the protein observed with Rictor (Protor) stabilizes the complex. The mammalian stress-activated protein kinase-interacting protein 1 (mSin1) is the regulator of mTORC2 activity. Both mTORC1 and mTORC2 are crucial in the integration of extracellular signals and energy status to control cell growth and cell survival [44]. 

When mTORC1 is activated, it promotes protein synthesis by phosphorylating the eukaryotic translation initiation factor 4E-BP1 and the ribosomal protein S6 kinase beta-1 (S6K1), enhancing the translation of mRNAs vital for growth and cell cycle progression. Furthermore, mTORC1 inhibits autophagy, a lysosomal degradation pathway activated during nutrient deficiency. Sestrins inhibit mTORC1 and therefore block cell cycle progression and proliferation [44]. mTORC2, less understood than mTORC1, enables cells to adapt and survive changes in energy intake. Sestrin-mediated mTORC1 downregulation activates mTORC2, aiding in stress adaptation [44].

In addition, Sestrins are involved in regulating autophagy/mitophagy, a cellular process that helps remove damaged organelles via activation of Unc-51-like protein kinase 1 (ULK1) [45,46,47]. Mitophagy specifically removes damaged mitochondria to restore the cellular redox balance and avoid oxidative stress [45,46,47]. 

The ER lumen supports protein maturation, involving tightly regulated folding and assembly of secretory and membrane-bound proteins. Disruptions in protein folding trigger an unfolded protein response (UPR) [48,49,50]. The UPR is regulated by transcription factors XBP1, ATF4, and ATF6. XBP1 and ATF6 can induce the expression of Sestrins, which, in turn, control protein synthesis and promote the degradation of misfolded proteins [42]. Notably, the Sestrin/mTORC1 pathway works alongside another pathway where PERK triggers cell cycle arrest through p53 [51].

The accumulation of cellular damage, particularly DNA damage, is responsible for the onset of diseases, including cancer. The p53/Sestrin/kinase axis protects against the accumulation of damage, thereby mitigating the development of malignancies [39].

## 2. p53/Sestrin Regulated Signaling Pathways

The p53/Sestrin/Kinase axis plays a significant role in preventing cellular stress and averting the accumulation of damage to macromolecules and organelles. Sestrins provide time for repair by modulating proliferation, protein synthesis, and metabolism. They initiate protective mechanisms that safeguard against damage (Figure 2).

### 2.1. p53 Target Genes Sestrin 1, Sestrin 2, and Sestrin 3 Link Cellular Stress to mTOR Signaling

The mTOR signaling pathway, vital for cell growth and survival, is downregulated during adverse conditions, such as nutrient scarcity, genotoxic stress, and hypoxia, which activate the tumor suppressor p53 [41,44,52]. This inhibition of mTOR signaling can occur through the p53-induced expression of Sestrin 1, 2, and 3, emphasizing their roles as negative regulators of cell growth and proliferation [15,27,40].

mTORC1 activation is primarily driven by Rheb, a small GTPase that directly binds to mTORC1 in its GTP-bound state. Additionally, a group of small GTPases known as RagA–D further enhances mTORC1 activation [44]. In their active state (RagA/B with GTP and RagC/D with GDP), the RagA/B:RagC/D complex binds to mTORC1, supporting its activation by Rheb. 

Cellular stress triggers the activation of p53, which subsequently induces the expression of its target genes, Sestrin 1, 2, and 3 [39,40]. Sestrins, in turn, induce the AMP-activated kinase (AMPK). Thereafter, AMPK phosphorylates and activates the tuberous sclerosis complex (TSC), an inhibitor of mTORC1 [39,42,44]. In addition, Sestrins can block GATOR1/GATOR2 which further suppresses mTORC1 activity [39,42,44,53,54]. Thus, p53/Sestrin-mediated inhibition of mTORC1 blocks cell growth. Inhibition of mTORC1 subsequently activates mTORC2, which plays a role in adapting to changes in energy supply and, therefore, promoting cell survival [39] (Figure 3).

### 2.2. The p53 Target Gene Sestrin 2 Controls Antioxidant Defense Mechanisms

As part of p53-activated antioxidant genes, Sestrins play a crucial role in suppressing reactive oxygen species (ROS), protecting against oxidative stress, preventing genomic instability, and inhibiting cellular transformation [38,40]. They achieve ROS control by promoting the expression of antioxidative proteins. Specifically, Sestrin 2 regulates the induction of an antioxidative response through the nuclear factor erythroid 2-related factor 2 (Nrf2) [36]. 

Sestrin 2 forms a complex with ULK1 and the autophagic cargo receptor p62/sequestosome-1 (SQSTM1). This complex facilitates the phosphorylation and degradation of p62/SQSTM1 and its associated substrate, Kelch-like ECH-associated protein 1 (Keap1), which is a major suppressor of Nrf2 [55]. Consequently, the stabilization of Nrf2 initiates an antioxidative response that safeguards cells from oxidative stress and the accumulation of cellular damage, thereby preventing the development of malignancies [56] (Figure 4).

Interestingly, Sestrin 2 possesses its own cysteine sulfinyl reductase activity, enabling the reduction and recycling of oxidized Peroxiredoxin. Consequently, Sestrin 2 plays a direct role in the elimination of highly reactive hydroperoxides and is thus an essential part of the oxidative response [38].

### 2.3. Sestrins: Initiators of Autophagy and Mitophagy

Autophagy is a lysosome-dependent degradation mechanism responsible for the targeted removal and recycling of cellular components such as proteins and organelles [57,58]. Initially identified as a survival response to starvation, autophagy is now recognized to be crucial for maintaining cellular homeostasis even in non-starved conditions [59].

Inhibition of mTORC1 by p53-induced Sestrin expression can activate ULK1. Once activated, ULK1 induces autophagy and phosphorylates the autophagy-related proteins 13 (Atg13), the focal adhesion kinase interacting protein of 200 kD (FIP200), and Atg101. These phosphorylation events serve as crucial triggers, initiating and activating lysosomal degradation by autophagosomes [60]. In this manner, damaged proteins and even entire organelles can be effectively removed, preventing the accumulation of cellular damage and reducing the risk of the development of malignancies, including cancers.

Mitophagy, a specialized form of autophagy, selectively degrades damaged or dysfunctional mitochondria. It is primarily regulated by PTEN-induced kinase 1 (PINK1) and Parkin [42]. Mitophagy maintains mitochondrial quality, prevents their accumulation, and adjusts mitochondrial numbers in response to metabolic changes and cellular developmental stages, such as differentiation [61]. 

Parkin is an E3 ubiquitin ligase that plays a crucial role in cellular quality control. Upon mitochondrial damage, Parkin is recruited and undergoes phosphorylation by PINK1. Following phosphorylation, Parkin attaches ubiquitin molecules to various substrates on the mitochondrial outer membrane. This ubiquitination process marks damaged mitochondria for degradation through mitophagy. Sestrin 2 plays a pivotal role in enhancing PINK1/Parkin-mediated mitophagy through two primary mechanisms. Firstly, Sestrin 2 interacts with ULK1, which leads to the phosphorylation of Beclin1 at Ser14. This phosphorylation event promotes the binding of Beclin1 to Parkin and further phosphorylates Parkin itself. These actions facilitate the translocation of Parkin to mitochondria, a critical step in mitophagy [42,62]. Secondly, Sestrin 2 can directly interact with Parkin, aiding in the translocation of Parkin to mitochondria and its subsequent accumulation within the mitochondria. These coordinated actions of Sestrin 2 enhance the efficiency of mitophagy, ensuring the removal of damaged mitochondria and the maintenance of cellular health [42,47] (Figure 5).

In summary, the p53 target Sestrin 2, in collaboration with the kinases PINK1 and ULK1, plays an essential role in removing damaged mitochondria. This process prevents the accumulation of ROS and, consequently, oxidative damage to macromolecules such as lipids, proteins, and DNA.

### 2.4. Sestrins and the UPR

The unfolded protein response (UPR) is a cellular stress response mechanism that monitors and manages the folding of proteins within the endoplasmic reticulum (ER). When proteins cannot fold correctly or accumulate within the ER, signaling pathways within the UPR are activated. These pathways aim to restore ER homeostasis by enhancing protein folding, degrading misfolded proteins, and reducing the synthesis of new proteins. However, if the stress is severe or prolonged, the UPR can trigger cell death. The UPR is crucial for maintaining cellular function and is implicated in various diseases [49,63,64].

The primary transcription factor inducing the UPR is XBP1, which can induce Sestrins, particularly Sestrin 2. Through Sestrin-mediated inhibition of mTORC1, protein synthesis is slowed down, allowing more time for proper protein folding. Additionally, the induction of autophagy helps to remove damaged organelles and proteins, another crucial aspect of the UPR. In this way, Sestrins support the prevention of the accumulation of misfolded proteins and ensure the survival of the cell [42,65,66].

## 3. Sestrins and Cancer

Sestrins are a group of evolutionarily conserved proteins that play important roles in various cellular processes, including those related to cancer [39].

Sestrins are often considered tumor suppressors because they help protect cells from various forms of stress that can lead to cancer development. Sestrins primarily function by inducing stress responses that protect the cell, particularly against oxidative stress. This induction occurs through the activation of Nrf2 via the kinase ULK1 (as described in Section 2.2). In cases where damage to macromolecules, such as DNA, proteins, and lipids, accumulates despite the initiation of anti-stress responses, Sestrins facilitate the allocation of time for repair mechanisms. This can be achieved either through inhibition of mTORC1 (as discussed in Section 2.1), by removal of damage from proteins, or the elimination of damaged organelles or mitochondria (as described in Section 2.3).

### 3.1. mTORC1 Triggers the Proliferation of Malignant Cells and Facilitates the Formation of Metastases

Under normal physiological conditions, the serine/threonine-kinase mTOR plays a central role in the regulation of cell growth and division. Conversely, in the context of cancer, aberrantly activated mTOR transmits signals that stimulate proliferation, metastasis, and infiltration of tumor cells into adjacent healthy tissues [67]. Among these, the PTEN/AKT/TSC pathway stands out as the primary activator of mTORC1. Mutations in genes within this pathway can give rise to malignant tumors [68,69]. In liver cancer, the activated mTORC1 pathway is implicated in tumor invasion and metastasis by increasing the expression of matrix metalloproteinase 9 [70]. Similarly, the mTOR signaling pathway has been identified as a regulator of the proliferation and survival of colon cancer stem cells. In sporadic colon cancer, colon cancer stem cells can contribute to recurrence and metastasis [71]. It has been demonstrated that extracellular growth signals can activate mTORC1. mTORC1, in turn, inhibits the activity of the ring finger protein 168 (RNF168) and promotes its degradation by phosphorylating serine 60 of RNF168. This process significantly reduces the ubiquitination modification of histone H2A and H2A histone family member X (H2AX) following DNA damage. Consequently, it impairs the DNA damage response and reduces genome stability, fostering malignant cell transformation and cancer [68,72]. Sestrins can inhibit mTORC1 via activation of TSC and inhibition of GATOR2 (as discussed in Section 2.1), substantially downregulating mTORC1 activity. Thus, Sestrins can impede the mTORC-dependent proliferation of malignant cells and block the formation of metastases.

### 3.2. Antioxidative Defense and Cancer Prevention

Sestrins can form a complex with the serine/threonine protein kinase ULK1, leading to the degradation of Keap1. Consequently, Nrf2 is released, initiating an antioxidative response (as discussed in Section 2.2; Figure 4). By inducing an antioxidative defense, ROS are scavenged, thereby preventing oxidations and associated damage to macromolecules such as DNA, proteins, or lipids. Specifically, the prevention of DNA damage inhibits the accumulation of mutations, consequently mitigating the development of malignancies. Thus, Sestrins protect the cell from degeneration through the induction of an antioxidative response [73]. As a result, Sestrins prevent the development of cancer.

However, the induction of an antioxidative response is a double-edged sword. Cancer cells distinguish themselves from normal cells through their elevated growth and proliferative capacity, often associated with Nrf2 overactivation. The reduced state of glutathione (GSH) is essential for cell proliferation due to its detoxification and antioxidant defense functions. Elevated activation of Nrf2 significantly enhances the transcription of the genes responsible for NADPH production, the primary cofactor in GSH synthesis [74], thus promoting tumor growth. In addition, the overactivation of Nrf2 in cancer cells leads to a substantial increase in the expression of metabolic enzymes that enhance glucose and glutamine metabolism within the pentose phosphate pathway. This augmentation supports the synthesis of purines and amino acids, collectively contributing to metabolic reprogramming in favor of cell proliferation [74]. In addition to their capacity for unlimited proliferation, cancer cells are also characterized by their ability to evade apoptosis. Cancer cells often exhibit high expression levels of ROS-scavenging enzymes, providing resistance to ROS-induced cell death. The heightened activation of Nrf2 counteracts the accumulation of ROS by upregulating antioxidative enzymes, thereby shielding cancer cells from the induction of apoptosis [73,75,76].

Hence, the Sestrin-mediated induction of Nrf2 displays two sides of the same coin. On the one hand, it prevents the accumulation of mutations, thereby blocking tumorigenesis. On the other hand, an established tumor benefits from Nrf2 activation, as it enhances its survival and shields it from apoptosis induction [73].

### 3.3. Sestrin-Induced Autophagy/Mitophagy

Autophagy and mitophagy are also double-edged swords. The housekeeping role of autophagy maintains the turnover of proteins and organelles, ensuring cellular homeostasis and health, thereby preventing disease conditions. In response to stress, autophagy is crucial in removing damaged proteins and organelles. This damage mitigation function of autophagy can be vital for the survival of tumor cells, which are frequently exposed to metabolic stress due to inadequate vascularization [77,78]. While more vulnerable to metabolic stress, cells deficient in autophagy display clear activation of the DNA damage response and an elevated frequency of chromosomal gains and losses. Therefore, by maintaining cellular quality control, autophagy may restrain genomic instability, thereby limiting tumor initiation and progression [78,79,80].

Sestrins induce autophagy/mitophagy through the serine/threonine kinases ULK1 and AMPK (as described in Section 2.3). On the one hand, this protects against accumulating damage to organelles, especially mitochondria. The latter prevents the generation of ROS and DNA damage due to oxidation. This, in turn, can prevent tumor formation. On the other hand, an established tumor offers an opportunity to adapt to altered nutrient conditions, promoting the survival of malignant cells [78].

## 4. Conclusions

Tumor suppressor p53 is activated in response to cellular stress [7,8]. p53 is responsible for the activation of Sestrins. The Sestrin family consists of three members: Sestrin 1, Sestrin 2, and Sestrin 3, with Sestrin 2 being the most extensively studied family member. The Sestrins perform essentially similar functions. The primary role of Sestrins is to initiate anti-stress responses within the cell. These anti-stress responses encompass a deceleration of cell proliferation and adaptation to fluctuations in nutrient availability. Furthermore, they involve the stimulation of antioxidant defense mechanisms, the promotion of autophagy and mitophagy, and safeguard the cell against stress by eliminating misfolded proteins [39]. The central core of these stress responses is formed by the serine/threonine kinases mTOR, ULK1, and AMPK. mTOR exists in two distinct complexes, known as mTORC1 and mTORC2, and it plays a critical role in cell proliferation and metabolic regulation. The mTORC1 complex is inhibited by Sestrins, resulting in a deceleration of cellular growth. The inhibition of mTORC1 concurrently increases the activity of the mTORC2 complex, enabling the cell to adapt its metabolism to fluctuations in nutrient availability [39]. AMPK and ULK1 are crucial in facilitating cellular adaptation to changing metabolic conditions. Furthermore, ULK1 is significantly involved in stabilizing Nrf2, which activates the antioxidative defense mechanisms to protect against the accumulation of oxidative damage, such as DNA damage [73]. Additionally, ULK1 and AMPK are involved in the induction of autophagy and mitophagy. This allows cells to adapt to inadequate nutrient supply and eliminate damaged organelles. The removal of dysfunctional mitochondria, in particular, prevents the release of reactive oxygen species (ROS) and, consequently, the accumulation of damage to macromolecules [39]. Sestrins are also activated by the accumulation of misfolded proteins in the ER. In this context, they provide the cell with the necessary time to eliminate the misfolded proteins through degradation by inhibiting mTORC1 [42,65,66].

The induction of such stress responses has a dual role in the context of cancer. On one hand, anti-stress responses prevent the formation and accumulation of cellular damage, thereby reducing the risk of cancer development. On the other hand, stress responses also facilitate the survival of tumor cells and complicate the treatment of cancer [39]. Thus, it is of utmost clinical relevance to understand these signaling pathways so they can be used strategically for cancer prevention or so they can be inhibited during cancer treatments, aiming for the optimal treatment outcome.

## Figures and Tables

**Figure 1 cancers-15-05390-f001:**
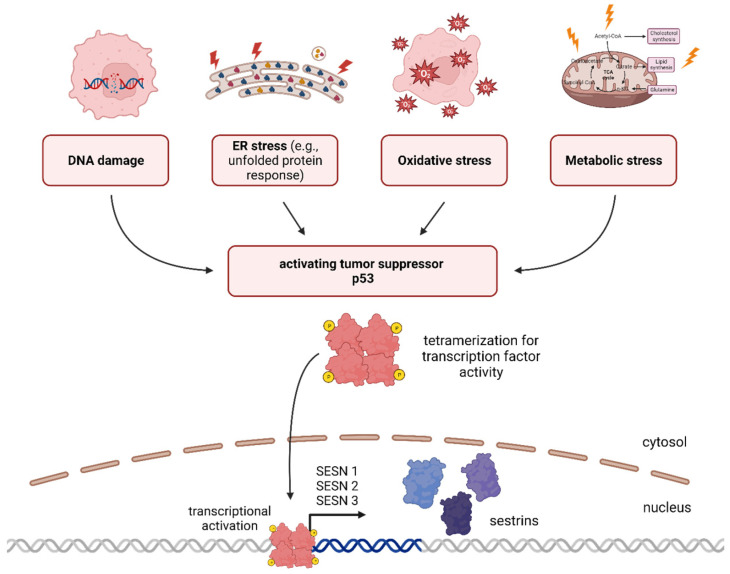
p53 is a tumor suppressor protein activated by cellular stressors, including DNA damage, ER stress, oxidative stress, and metabolic stress. The Sestrin (SESN) family comprises Sestrin 1, 2, and 3, which are target genes of p53. The graphic was created using BioRender (www.biorender.com).

**Figure 2 cancers-15-05390-f002:**
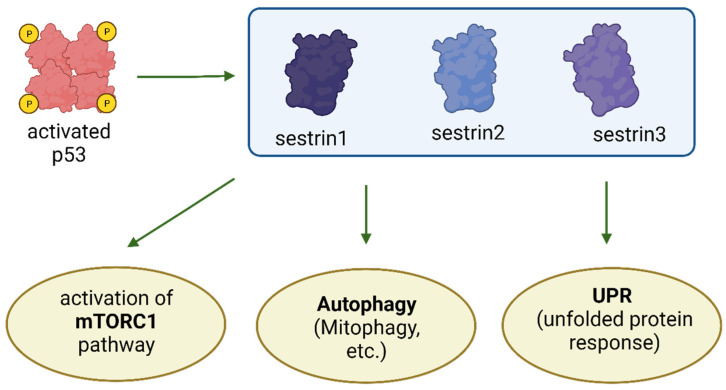
Activated p53 stimulates the expression of Sestrins. Sestrins, in turn, induce anti-stress responses, including the deceleration of cell proliferation and adaptation to changing nutrient conditions through mTORC1 inhibition, the promotion of autophagy/mitophagy for organelle quality control, and the initiation of an unfolded protein response (UPR) to prevent the accumulation of misfolded proteins. The graphic was created using BioRender (www.biorender.com).

**Figure 3 cancers-15-05390-f003:**
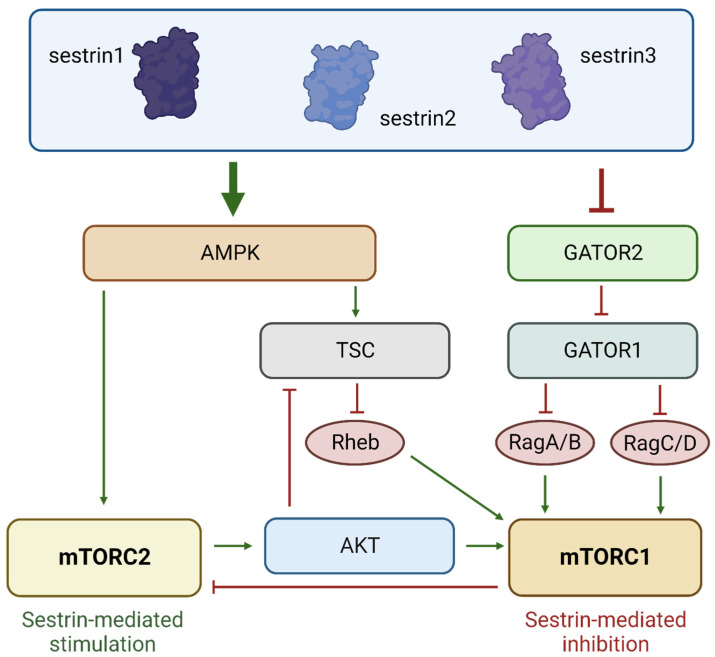
Sestrins activate AMPK, and AMPK phosphorylates and activates TSC, which, in turn, inhibits mTORC1. Additionally, Sestrins can also inhibit GATOR2, resulting in the inhibition of mTORC1 through GATOR1. The inhibition of mTORC1 and activation of AMPK can further increase the activity of mTORC2. The inhibition of mTORC1 slows down cell proliferation, while the activation of mTORC2 allows the cell to adapt to changing nutrient conditions. The graphic was created using BioRender (www.biorender.com).

**Figure 4 cancers-15-05390-f004:**
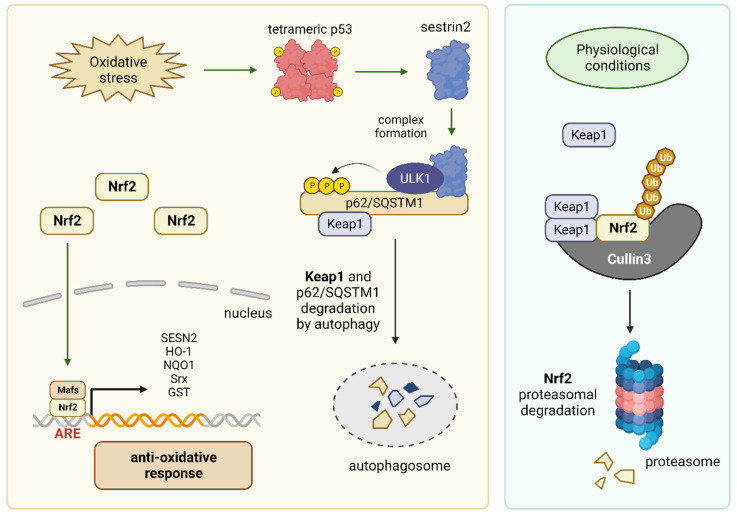
(**Left panel**): Oxidative stress can induce p53 activation, which in turn triggers the activation of Sestrins. Sestrins subsequently promote the degradation of Keap1 via ULK, resulting in the accumulation and activation of Nrf2 and the induction of an antioxidative stress response. (**Right panel**): In the absence of oxidative stress, Nrf2 is bound to Keap1, which triggers the degradation of Nrf2. The graphic was created using BioRender (www.biorender.com).

**Figure 5 cancers-15-05390-f005:**
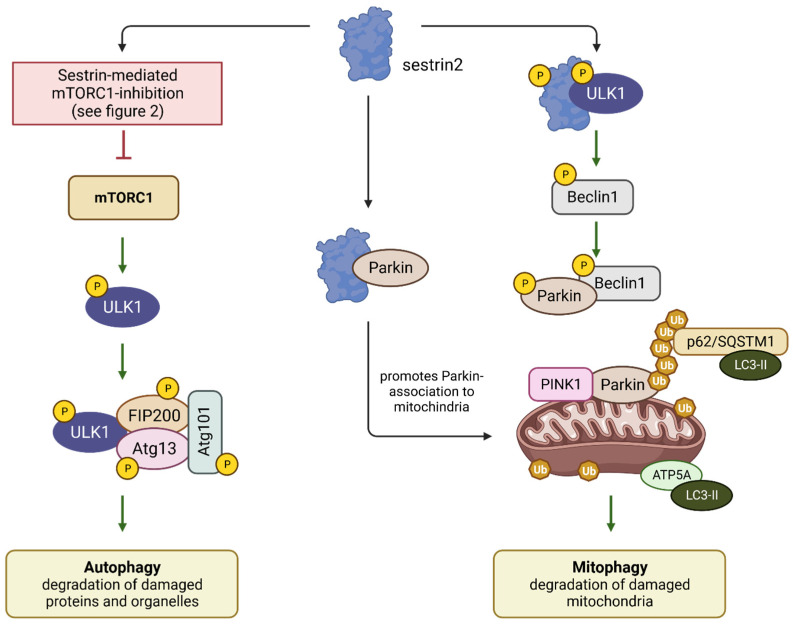
Sestrins inhibit mTORC1, thereby enabling the activation of the kinase ULK1. This activation leads to the assembly of a complex consisting of phosphorylated ULK1, FIP200, Atg13, and Atg101, initiating the process of autophagy. ULK1 can also phosphorylate Beclin1, which forms a complex with Parkin. This complex relocates to the mitochondria, where, in conjunction with PINK1, it triggers mitophagy, a specific form of autophagy. Mitophagy is responsible for eliminating damaged mitochondria, thereby preventing malfunctions and excessive ROS production. The graphic was created using BioRender (www.biorender.com).

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
