# Peer review of "The Important Role of Protein Kinases in the p53 Sestrin Signaling Pathway"

_cancers, 2023, doi:10.3390/cancers15225390_

Round 1

Reviewer 1 Report

Comments and Suggestions for Authors

This is an excellent review that describes the function of the sestrins, with a particular focus on sestrin2. It is quite short but very informative. The figures are perfectly done using the Biorender program. The role of sestrins as p53 targets is described in detail with a focus on ROS and mitophagy. However, I do have some critical comments:

- As mentioned above, a considerable part of the review is on sestrin2, which needs to be reflected in the abstract and conclusions.

- There are typos and some inconsistencies like Parkin is sometimes written Parkin and sometimes Parkin and many others. This needs to be corrected.

- Some of the first discoveries on sestrins need to be highlighted, especially: doi: 10.1126/science.1095569.PMID: 15105503

-Using one color for sestrin2 and the other ones through the figures would have been of advantage

Author Response

Answers to the comments of Reviewer #1:

Comment of Reviewer #1:

This is an excellent review that describes the function of the sestrins, with a particular focus on sestrin2. It is quite short but very informative. The figures are perfectly done using the Biorender program. The role of sestrins as p53 targets is described in detail with a focus on ROS and mitophagy. However, I do have some critical comments:

Answer:

We appreciate the reviewer for his/her constructive feedback. We have thoroughly revised the manuscript and addressed the raised critics (all changes in the text are highlighted in yellow).

Comment of Reviewer #1:

As mentioned above, a considerable part of the review is on sestrin2, which needs to be reflected in the abstract and conclusions.

Answer:

We concur with the reviewer and have provided detailed elaboration on this matter, subsequently revising the text. We would like to emphasize that in both the abstract and conclusion, we have highlighted Sestrin 2 as the most extensively investigated member of the Sestrin family (please see page 1 and 15). In the newly added section 'Simple Summary,' we also explicitly emphasize that all Sestrins are addressed in this review, with a specific focus on Sestrin 2 (see page 1).

Comment of Reviewer #1:

There are typos and some inconsistencies like Parkin is sometimes written Parkin and sometimes Parkin and many others. This needs to be corrected.

Answer:

We appreciate the reviewer's feedback. We have thoroughly reviewed orthography and grammar using a spelling tool, rectifying all spelling errors, including ensuring consistent spelling of 'Parkin (see page 9).

Comment of Reviewer #1:

Some of the first discoveries on sestrins need to be highlighted, especially: doi: 10.1126/science.1095569.PMID: 15105503

Answer:

We enclosed the recommended paper, along with three supplementary studies that detail the initial identification of Sestrins (pages 4/5 and throughout the following text; Ref. 35-38).

The new References are:

Peeters, H.; Debeer, P.; Bairoch, A.; Wilquet, V.; Huysmans, C.; Parthoens, E.; Fryns, J.P.; Gewillig, M.; Nakamura, Y.; Niikawa, N.; et al. PA26 is a candidate gene for heterotaxia in humans: identification of a novel PA26-related gene family in human and mouse. Hum. Genet. 2003, 112, 573–580, doi:10.1007/s00439-003-0917-5.

Budanov, A.V.; Shoshani, T.; Faerman, A.; Zelin, E.; Kamer, I.; Kalinski, H.; Gorodin, S.; Fishman, A.; Chajut, A.; Einat, P.; et al. Identification of a novel stress-responsive gene Hi95 involved in regulation of cell viability. Oncogene 2002, 21, 6017–6031, doi:10.1038/sj.onc.1205877.

Velasco-Miguel, S.; Buckbinder, L.; Jean, P.; Gelbert, L.; Talbott, R.; Laidlaw, J.; Seizinger, B.; Kley, N. PA26, a novel target of the p53 tumor suppressor and member of the GADD family of DNA damage and growth arrest inducible genes. Oncogene 1999, 18, 127–137, doi:10.1038/sj.onc.1202274.

Budanov, A.V.; Sablina, A.A.; Feinstein, E.; Koonin, E.V.; Chumakov, P.M. Regeneration of peroxiredoxins by p53-regulated sestrins, homologs of bacterial AhpD. Science 2004, 304, 596–600, doi:10.1126/science.1095569.

Comment of Reviewer #1:

Using one color for sestrin2 and the other ones through the figures would have been of advantage

Answer:

Once more, we concur with the reviewer. We have updated the figures, ensuring consistent colors for representing Sestrins in our diagrams (Figures 1-4).

Thank you for helping us to impove our manuscript.

Reviewer 2 Report

Comments and Suggestions for Authors

This is very nice written and organized review, dedicating to important signal transduction pathway.  proteins described in the review are, indeed, critical players in cell response to different kinds of stress. The role of Sestrins in cancer and other pathological conditions is well mentioned. Illustrations are helpful and well designed.  References are well selected.

Author Response

Answers to the comments of Reviewer #2:

Comment of Reviewer #2:

This is very nice written and organized review, dedicating to important signal transduction pathway.  proteins described in the review are, indeed, critical players in cell response to different kinds of stress. The role of Sestrins in cancer and other pathological conditions is well mentioned. Illustrations are helpful and well designed.  References are well selected.

Answer:

Thank you for the fast review process and the highly positive feedback.